# The clinicopathological significance of Thrombospondin-4 expression in the tumor microenvironment of gastric cancer

Kenji Kuroda[1,2,3], Masakazu Yashiro [1,2,3]*, Tomohiro Sera[1,2,3], Yurie Yamamoto[2,3], Yukako Kushitani[1,2,3], Atsushi Sugimoto[1,2,3], Syuhei Kushiyama[1,2,3], Sadaaki Nishimura[1,2,3], Shingo Togano[1,2,3], Tomohisa Okuno[1,2,3], Tatsuro Tamura[1], Takahiro Toyokawa[1], Hiroaki Tanaka[1], Kazuya Muguruma[1], Masaichi Ohira[1]

1 Department of Gastroenterological Surgery, Osaka City University Graduate School of Medicine, Osaka city, Japan, 2 Molecular Oncology and Therapeutics, Osaka City University Graduate School of Medicine, Osaka city, Japan, 3 Cancer Center for Translational Research, Osaka City University Graduate School of Medicine, Osaka city, Japan

* m9312510@med.osaka-cu.ac.jp

**Data Availability Statement:** All relevant data are within the paper and its Supporting Information files.

## Abstract

### Introduction

Thrombospondin-4 [1] is an extracellular glycoprotein involved in wound healing and tissue remodeling. Although THBS4 is reportedly frequently expressed in solid tumors, there are few reports of the clinicopathological features of carcinomas with THBS4 expression. We evaluated the clinicopathologic significance of THBS4 expression in gastric carcinoma (GC).

### Materials and methods

We retrospectively analyzed the cases of 584 GC patients. The expression of THBS4 in each tumor was evaluated by immunohistochemistry. We then divided the patients into the THBS4-high (n = 223, 38.2%) group and THBS4-low (n = 361, 61.8%) group. THBS4 expression in cancer-associated fibroblasts (CAFs), normal-associated fibroblasts (NFs) and gastric cancer cell lines was examined by western blotting.

### Results

THBS4 is expressed on stromal cells with αSMA or Podoplanin expression in the GC microenvironment, but not expressed on cancer cells with cytokeratin expression. The western blot analysis results showed that CAFs (but not NFs and cancer cells) expressed THBS4. Compared to the THBS4-low expression status, the THBS4-high expression status was correlated with higher αSMA expression, higher invasion depth, lymph-node metastasis, lymphatic invasion, peritoneal cytology, peritoneal metastasis, larger tumor size, microscopic diffuse type, and the macroscopic diffuse infiltrating type. The THBS4-high group's 5-year overall survival rate was significantly poorer than that of the THBS4-low group. A multivariate analysis revealed that THBS4 expression was an independent prognostic factor.

**Funding:** This study is partially supported by Japan Society for the Promotion of Science KAKENHI (Grant-in-Aid for Scientific Research B; Grant Number JP18H02883 (https://www.jsps.go.jp/english/e-grants/) to MY. The funder had no role in study design, data collection and analysis, decision to publish, or preparation of the manuscript. There was no additional external funding received for this study.

**Competing interests:** The authors have declared that no competing interests exist.

## Conclusion

THBS4 is expressed on CAFs in the gastric cancer microenvironment. THBS4 from CAFs is associated with the metastasis of cancer cells, and is a useful prognostic indicator for gastric cancer patients.

## Introduction

Despite recent advances in diagnostic techniques and therapies for gastric cancer (GC), the prognosis of GC remains poor; GC is the fifth most common fatal carcinoma and the third leading morbidity in the world [2, 3]. To improve the prognosis of GC patients, new strategies based on its biological behavior are necessary. Tumor progression has been recognized as the product of evolving crosstalk between the cancer cells and the surrounding tissue or tumor stroma[4]. The elucidation of the characteristic features of tumor stroma could provide a new biological marker in the crosstalk.

Thrombospondin-4 (THBS4) is one of the extracellular secreted glycoproteins involved in wound healing and tissue remodeling by regulating the organization, repair, and remodeling of the extracellular matrix [5–7]. It was recently reported that THBS4 is frequently expressed in the tumor stroma of some types of solid cancers such as breast cancer and prostate cancer [8] [9]. By conducting a transcriptome-wide comparative analysis, Forester et al. [10] observed that THBS4 is a potent marker for diffuse-type gastric cancer, and they detected THBS4 expression in tumor stroma which showed positivity for α-Smooth muscle actin (αSMA) and vimentin.

Although it has been reported that THBS4 is highly expressed in cancer stroma, the significance of THBS4 in tumor stroma in the development of gastric cancer has been controversial. THBS4 was reported to stimulate the development of cancer [8, 10–13], whereas other studies found that THBS4 acts as a tumor suppressor [14, 15]. Few investigations have examined the clinicopathological significance of THBS4, and we conducted the present study to evaluate the clinicopathological significance of THBS4 expression in gastric cancer, especially in tumor stroma.

## Materials & methods

### Patients and clinical materials

This study was retrospective analysis of 585 gastric cancer patients who received gastrectomy at Osaka City University Hospital. Gastric cancer tissues were obtained from each patient. The pathological diagnoses and classifications were made according to the Japanese classification of gastric carcinoma (14th edition) [15]. This study was approved by the Osaka City University Ethics Committee (approval number 924). Written informed consent for research was obtained from patients.

### Immunohistochemical determination of THBS4, Podoplanin, αSMA and Cytokeratin

The immunohistochemical determination of THBS4, Podoplanin, αSMA and cytokeratin were examined as the manufacturer's instructions. Shortly, slides were deparaffinized and activated by heating. After blocking endogenous peroxidase activity, the samples were incubated with anti-human THBS4 (R&D Systems, Minneapolis, MN: MAB2390; 1:200), anti-Pan-

Keratin (Proteintech, Rosemont, IL: 26411-1-AP; 1:3000), anti-Podoplanin (Santa Cruz Biotechnology, Dallas, TX: sc-376695; 1:300) or anti-human αSMA (Leica Biosystems Newcastle, Newcastle, UK: NCL-L-SMA; 1:200). The samples were incubated with biotinylated second antibody. The samples were treated with streptavidin-peroxidase reagent, and counterstaining with Mayer's hematoxylin. THBS4 expression was evaluated by intensity of staining and percentage of stained stromal cells, respectively: intensity was given scores 0–3 (0 = no, 1 = weak, 2 = moderate, 3 = intense), and the percentage of immunopositive stromal cells in all stroma cells was given scores 0–3 (0 = 0%, 1 = 1%-10%, 2 = 11%-50%, 3 = 51%-100%). The two scores were added to obtain the final result of 0–6. Expressions were considered THBS4-high when scores were 3 or more and THBS4-low when scores were 2 or less. Expressions were considered αSMA-high when scores were 5 or more and αSMA-low when scores were 4 or less. Two double-blinded independent observers who were unaware of clinical data and outcome evaluated, and when a discrepant evaluation between the two independent observers was found, the result was rechecked and discussed.

## Gastric fibroblast cell lines and cancer cell lines

Three pairs of cancer associated fibroblast (CAF82, CAF105) and normal tissue fibroblast (NF82, NF105) were established from tumoral gastric wall and non-tumoral gastric wall, respectively. CAF82 and NF82, CAF105 and NF105 were from same patients. These fibroblasts were used from third passage to fifth passage in culture. CAFs and NFs were examined by αSMA staining, as previously reported [16]. MKN45 and NUGC4, which are diffuse and intestinal type gastric cancer cell lines, respectively, were used.

## Immunostaining of fibroblasts

Fibroblasts were incubated into Lab-Tek II Chamber Slide System (Nunc, Naperville, IL) for 3 days. After washing with PBS and fixing with methanol for 10 min, fibroblasts were incubated with anti-podoplanin antibody (Santa Cruz Biotechnology, Dallas, TX: sc-376695; 1:300). The samples were incubated with biotinylated second antibody.

## Western blot analysis

Cell lysates were collected after different treatments. After the protein concentration of each sample was adjusted, electrophoresis was carried out using 10% Tris/Gly gels (Invitrogen, Inc., Gaithersburg, MD). The protein bands obtained were transferred to an Immobilon-P Transfer membrane (Amersham, Aylesbury, UK). The membrane was kept in PBS-T (10 mM PBS and 0.05% Tween 20) supplemented with 5% bovine albumin (Sigma, St. Louis, MO) at room temperature for 1 h. Then, the membrane was placed in PBS-T solution containing each primary antibody: THBS4 (R&D Systems), β-actin (1:1000; Cell Signaling), and allowed to react at room temperature for 2 h. The levels of specific proteins in each lysate were detected by enhanced chemiluminescence using ECL plus (Amersham) followed by autoradiography.

## Statistical analysis

Statistical analysis was performed using R for Mac OS X (version 3. 5. 2). The chi-square test was used to determine the significance of the difference between the covariates. Survival was measured from the date of surgery. Overall survival was calculated using Kaplan-Meier method, and survival curves were compared by log-rank test. The Cox proportional hazards model was used for multivariate analysis. A p-value $<0.05$ was defined as being statistically significant.

## Results

### Immunostaining findings of THBS4

Fig 1 provides representative immunostaining patterns of THBS4. THBS4 was stained at the cytoplasm and the cell membrane of stroma cells, but cancer cells with cytokeratin expression did not express THBS4. Whereas, the THBS4 expression of the stromal cells was in agreement with the cells showing alpha-smooth muscle actin (αSMA) or Podoplanin expression. Of the total of 584 cases, 223 (38.2%) were THBS4-high.

### The relationship between clinicopathological features and the THBS4 expression of stromal cells

The clinicopathological characteristics of all cases according to their THBS4 expression in stroma are summarized in Table 1. Compared to the THBS4-low expression in stroma, THBS4-high expression in stroma was significantly associated with higher αSMA expression, macroscopic type 4 (p<0.001), greater tumor diameter (p<0.001), diffuse-type histology (p<0.001), microscopic undifferentiated-type, higher tumor invasion (p<0.001), lymph node metastasis (p<0.001), lymphatic invasion (p<0.001), peritoneal cytology (p<0.001), and peritoneal metastasis (p<0.001). Among Borrmann's type 4 cancer of 61 cases, 52 (85.2%) were THBS4-high. The results of subgroup analysis for type-4 or other type gastric cancer were shown in S1 and S2 Tables.

### Survival outcomes

The 5-year overall survival rate of the patients in THBS4-high expression group (38.5%) was significantly poorer compared to that of the THBS4-low group (81.4%) (p<0.001; Fig 2). The results of subgroup analysis for type-4 or other type gastric cancer were shown in S1 Fig. Our analysis for each tumor stage revealed that the overall survival of the THBS4-high expression patients at Stage I and Stage III was significantly poorer than that of the patients with THBS4-low expression (p<0.001 and p = 0.001, respectively; Fig 2). The recurrence site according to pathological stages is shown in S2 Table. Hematogenous recurrence was frequent in patients at stage II patients, in compared with those at stage III.

### Univariate and multivariate analyses

Table 2 provides the results of the univariate and multivariate analyses for overall survival. The univariate analysis revealed that poor survival was significantly correlated with high αSMA expression (p<0.001), high THBS4 expression (p<0.001), age ≥ 65 years old (p<0.001), macroscopic type 4 (p<0.001), microscopic undifferentiated type (p<0.001), high T stage (p<0.001), lymph node metastasis (p<0.001), lymphatic invasion (p<0.001), vascular invasion (p<0.001), ascites cytology positive (p<0.001), peritoneal metastasis (p<0.001), and hepatic metastasis (p<0.001).

The multivariate analysis revealed that THBS4 expression, age ≥65 years, macroscopic type 4 (p<0.001), lymph node metastasis (p<0.001), cytology-positive status (p<0.001), peritoneal metastasis (p<0.001), and hepatic metastasis (p<0.001) were independent prognostic factors.

### THBS4 expression on CAFs and NFs

Fig 3A shows established CAFs and NFs stained with Podoplanin. The results of the western blot analysis revealed that the CAF82 and CAF105 significantly expressed THBS4 compared

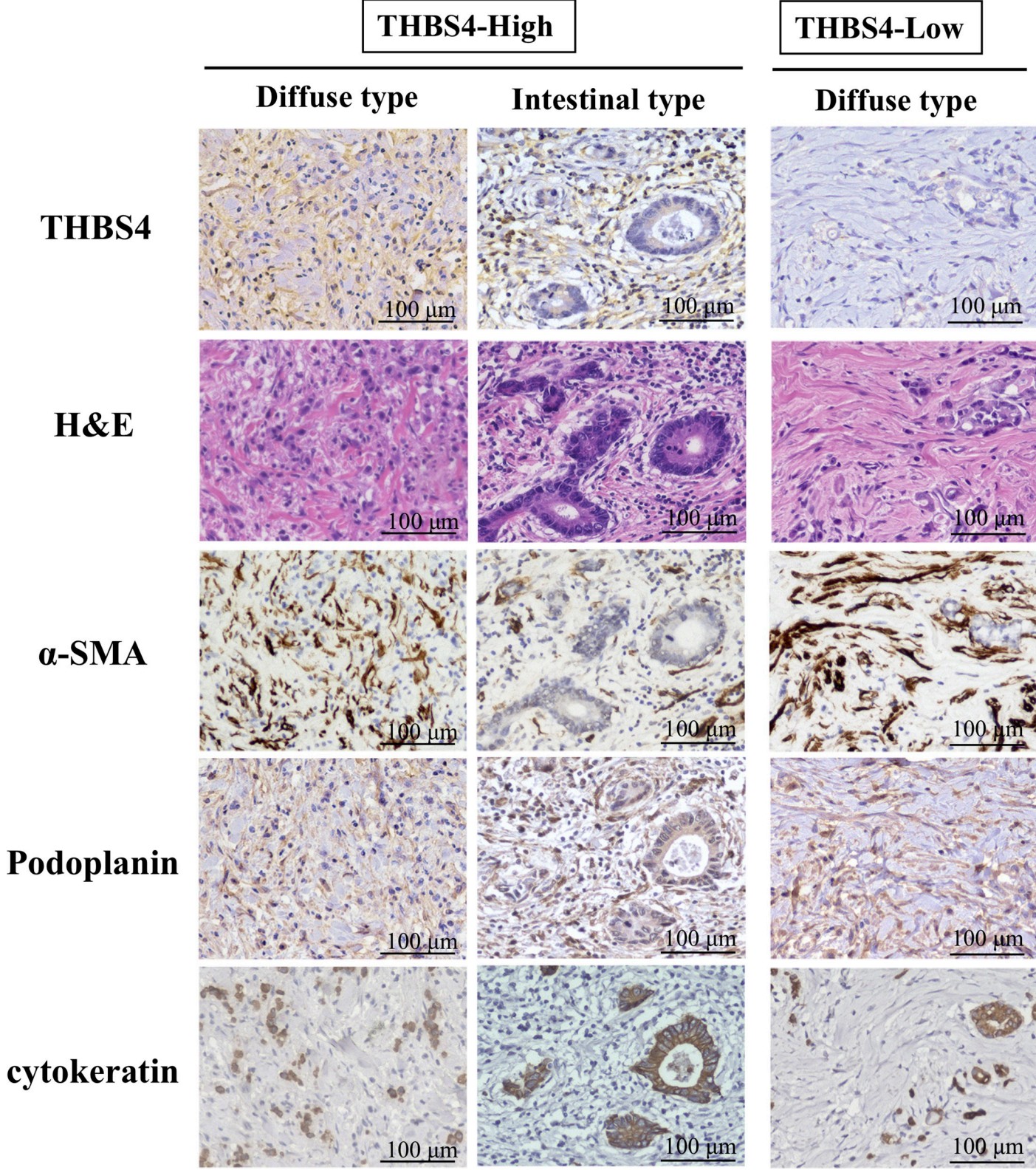

**Fig 1. Representative picture of THBS4 expression, H&E staining, αSMA expression, Podoplanin expression, and cytokeatin expression in diffuse and intestinal type of gastric cancer.** THBS4 was stained at the cytoplasm and cell membrane of stroma cells. The stromal cells with THBS4 expression were much to the cells showing αSMA or Podoplanin expression. cytokeratin was stained in cancer cells.

**Table 1. Correlation between the expression of THBS4 in tumor stromal cells and clinicopathologic features in 584 patients with gastric carcinoma.**

|  | THBS4 | | |
|---|---|---|---|
|  | **High (n = 223, 38.2%)** | **Low (n = 361, 61.8%)** | **p value** |
| Age (year-old) |  |  |  |
| < 65 | 105 (47.1%) | 151 (41.8%) |  |
| ≥ 65 | 118 (52.9%) | 210 (58.2%) | 0.247 |
| Gender |  |  |  |
| Female | 97 (43.5%) | 162 (44.9%) |  |
| Male | 126 (56.5%) | 199 (55.1%) | 0.810 |
| αSMA expression |  |  |  |
| High | 139 (62.3%) | 136 (37.7%) |  |
| Low | 84 (37.7%) | 225 (62.3%) | <0.001 |
| Macroscopic type |  |  |  |
| type 4 | 52 (23.3%) | 9 (2.5%) |  |
| Other types | 171 (76.7%) | 352 (97.5%) | <0.001 |
| Tumor diameter |  |  |  |
| < 50 | 103 (46.4%) | 250 (69.3%) |  |
| ≥ 50 | 120 (53.6%) | 111 (30.7%) | <0.001 |
| Microscopic type |  |  |  |
| Differentiated | 81 (36.3%) | 205 (56.8%) |  |
| Undifferentiated | 142 (63.7%) | 156 (43.2%) | <0.001 |
| Depth of tumor invasion |  |  |  |
| T1-2 | 73 (32.7%) | 264 (73.1%) |  |
| T3-4 | 150 (67.3%) | 97 (26.9%) | <0.001 |
| Lymph node metastasis |  |  |  |
| N0 | 83 (37.2%) | 246 (68.1%) |  |
| N1-3 | 140 (62.8%) | 115 (31.9%) | <0.001 |
| Lymphatic invasion |  |  |  |
| Absent | 56 (25.1%) | 204 (56.5%) |  |
| Present | 167 (74.9%) | 157 (43.5%) | <0.001 |
| Venous invasion |  |  |  |
| Absent | 176 (84.2%) | 305 (79.1%) |  |
| Present | 47 (15.8%) | 56 (20.9%) | 0.109 |
| Ascites cytology |  |  |  |
| Negative | 174 (78.0%) | 346 (95.8%) |  |
| Positive | 49 (22.0%) | 15 (4.2%) | <0.001 |
| Peritoneal metastasis |  |  |  |
| Absent | 200 (89.7%) | 349 (96.7%) |  |
| Present | 23 (10.3%) | 12 (3.3%) | 0.001 |
| Hepatic metastasis |  |  |  |
| Negative | 216 (96.9%) | 354 (98.1%) |  |
| Positive | 7 (3.1%) | 7 (1.9%) | 0.409 |
| pStage |  |  |  |
| I, II | 99 (44.4%) | 285 (78.9%) |  |
| III, IV | 124 (55.6%) | 76 (21.1%) | <0.001 |

with NF82 and NF105, respectively (t-test, p<0.001 and p<0.001, respectively). MKN45 and NUGC4 did not. (Fig 3B).

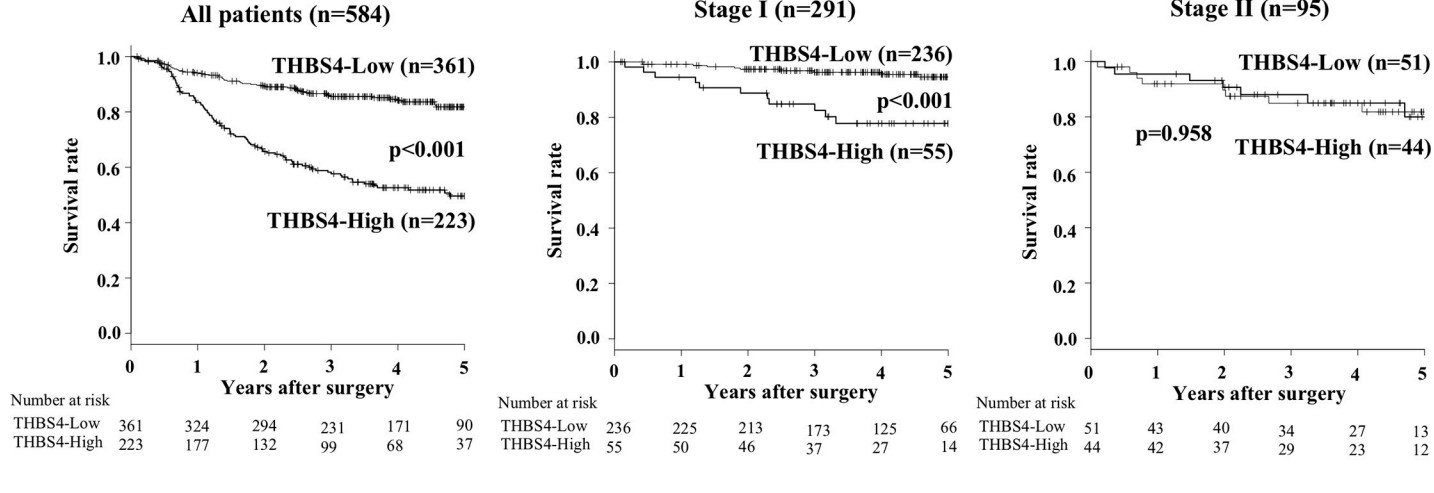

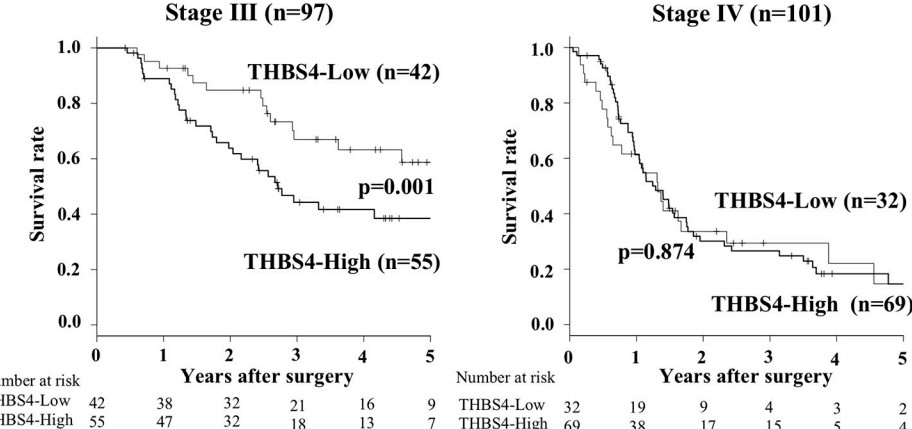

**Fig 2. The overall survival (OS) of the gastric cancer patients based on the THBS4 expression in their tumor stromal cells.** (A) The Kaplan-Meier survival curve indicates that the OS of all patients with high THBS4 expression in stromal cells was significantly worse than that of the patients with low THBS4 expression (p<0.001). (B–E) The Kaplan-Meier survival curve for each stage. The 5-year OS rates of the Stage I patients with high THBS4 expression were poorer than those of the Stage I patients with low THBS4 expression (p<0.001), and the same was true of the Stage III patients (p = 0.001).

## Discussion

THBS4 was reported to be expressed not only in normal tissues but also in some types of solid cancers [8, 10, 17]. In the present study, we observed that gastric tumors (especially the stromal cells in tumor microenvironment) also express THBS4. In addition, the stromal cells expressing THBS4 also expressed αSMA or podoplanin, which is a marker of CAFs. In fact, the western blotting indicated that THBS4 was expressed from CAFs, but not from NFs and cancer cell lines. In case analysis, THBS4-high expression status was correlated with high αSMA expression. These findings suggest that in gastric cancer, THBS4 may be expressed by the gastric cancer-associated fibroblasts.

We reported that CAFs play an important role in the progression, growth, and spread of gastric cancer [18]. In present study, we showed in the results that THBS4-high expression in stroma was significantly associated with higher αSMA expression, higher tumor invasion, lymph node metastasis, lymphatic invasion, peritoneal cytology, and peritoneal metastasis. We also found that high THBS4 expression in tumor stroma was significantly poorer than that of the patients with low THBS4 expression. Moreover, multivariate analysis for overall survival

**Table 2. Univariate and multivariate Cox multiple regression analysis with respect to overall survival after surgery in 584 patients with gastric carcinoma.**

|  | Univariate analysis | | Multivariate analysis | |
|---|---|---|---|---|
|  | Hazard Ratio (95% CI) | p value | Hazard Ratio (95% CI) | p value |
| THBS4-high | 3.43 (2.48–4.75) | <0.001 | 1.53 (1.05–2.25) | 0.028 |
| Age ≥65 year-old | 1.61 (1.16–2.23) | 0.004 | 1.74 (1.23–2.46) | 0.002 |
| Female (vs. Male) | 1.29 (0.93–1.79) | 0.123 |  |  |
| αSMA-high | 2.37 (1.70–3.31) | <0.001 | 1.27 (0.90–1.80) | 0.180 |
| Borrmann's type 4 | 7.41 (5.22–10.5) | <0.001 | 1.62 (1.05–2.50) | 0.042 |
| Tumor diameter ≥50 mm | 6.27 (4.37–9.00) | <0.001 | 1.49 (0.90–2.46) | 0.088 |
| Undifferentiated type | 1.72 (1.24–2.37) | 0.001 | 1.18 (0.81–1.70) | 0.386 |
| T3&4 (vs. T1&2) | 6.72 (4.61–9.80) | <0.001 | 1.41 (0.82–2.40) | 0.211 |
| N1-3 (vs. N0) | 8.29 (5.50–12.5) | <0.001 | 2.79 (1.62–4.80) | <0.001 |
| Lymphatic invasion | 5.22 (3.42–7.95) | <0.001 | 1.06 (0.61–1.83) | 0.837 |
| Vascular invasion | 3.10 (2.22–4.32) | <0.001 | 1.06 (0.73–1.53) | 0.772 |
| Cytology positive | 7.42 (5.23–10.5) | <0.001 | 1.91 (1.28–2.85) | 0.001 |
| Peritoneal metastasis | 9.11 (5.97–13.9) | <0.001 | 2.60 (1.64–4.12) | <0.001 |
| Hepatic metastasis | 5.90 (3.18–11.0) | <0.001 | 3.64 (1.89–7.02) | <0.001 |

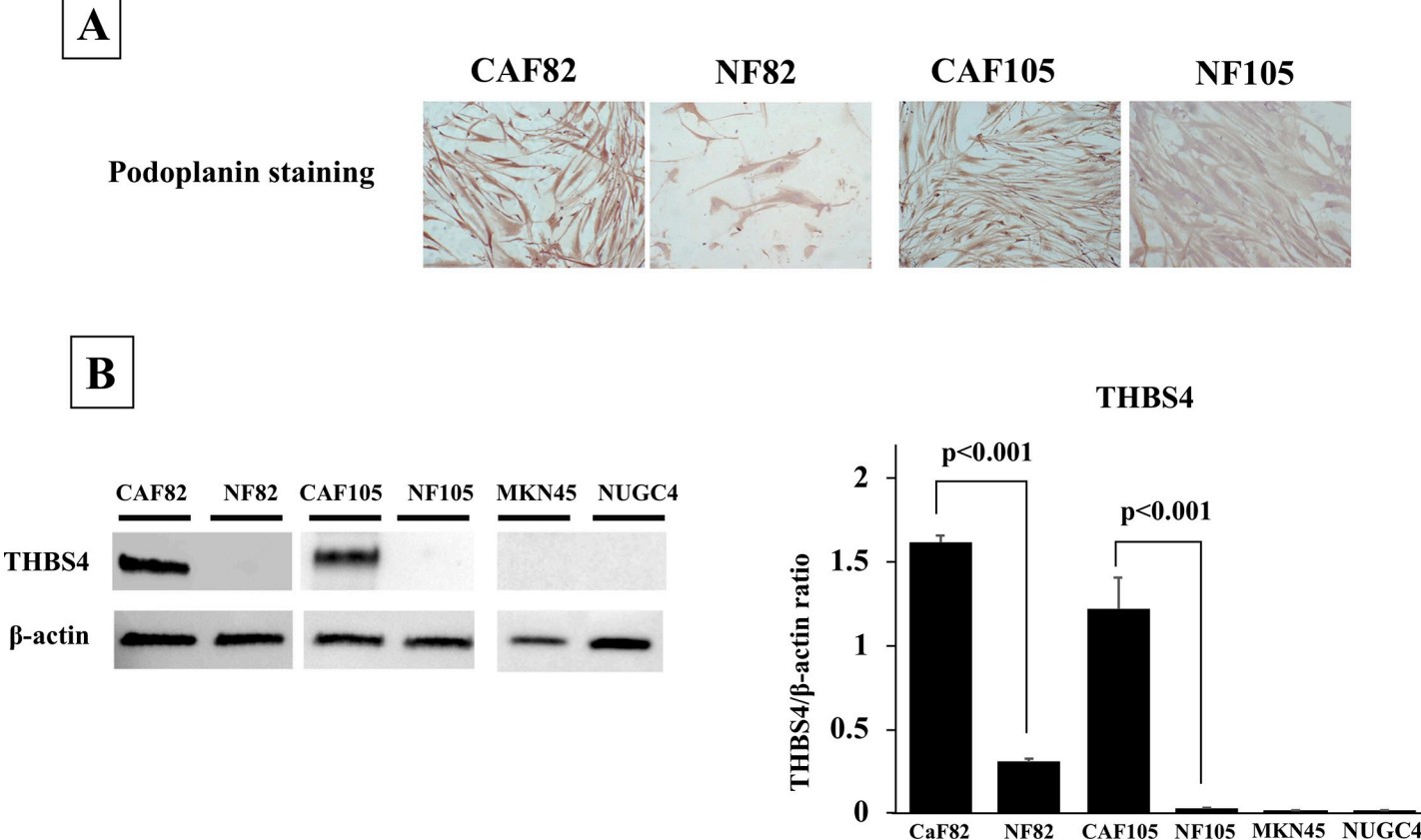

**Fig 3. The picture of CAFs and NFs stained with Podoplanin and the western blot analysis of THBS4. (A), Representative picture of Podoplanin staining.** Podoplanin was mainly stained at the cytoplasm fibroblasts. The expression level of Podoplanin was higher in CAFs, in compared with that in NFs. **(B), THBS4 expression.** CAFs expressed THBS4, but NFs and cancer cells did not.

showed that not αSMA expression on CAFs but THBS4 expression on CAFs was an independent prognostic factor. These findings indicated that THBS4 from CAFs might be associated with the invasion of cancer cells, and THBS4 from CAFs is associated with the metastasis of cancer cells.

By targeting the protein coding gene Kruppel-like factor 9 (KLF9) using gastric cancer cell lines, Chen et al. observed that THBS4 had positive effects on gastric cancer cells' proliferation and metastasis [1]. However, little is known about the molecular mechanism of THBS4 remains unclear. Muppala S et al. reported that THBS4 is upregulated in response to TGF-β and promotes the effect of TGF-β on angiogenesis, mediating cell proliferation and survival [10]. Forester et al. showed *THBS4* transcription on CAFs is stimulated by incubation with conditioned medium derived from human diffuse gastric cancer lines [4]. It is speculated that TGF-β derived from cancer cells stimulated THBS4 expression on CAFs resulting in angiogenesis and cancer progression.

In particular, the stage I and stage III patients with high THBS4 expression had significantly poorer prognosis than the stage I and III patients with low THBS4 expression, respectively. High THBS4 expression in stromal cells was revealed as an independent prognostic factor for gastric cancer patients. THBS4 from CAFs might be a useful prognostic indicator for gastric cancer patients, especially for those with stage I or III cancer.

In contrast, no significant difference in prognosis was shown between the THBS4-high group and THBS4-low group at stage II. Liver metastasis is a relatively frequent recurrence pattern among patients with stage II gastric cancer [19], and THBS4 expression is not correlated with vascular invasion or liver metastasis, which might be one of the reasons for the absence of a difference in the prognosis between our two THBS4 expression groups at stage II. Additionally, effects of adjuvant chemotherapy might be caused especially at stage II. It was difficult to compare adjuvant regimen according to the stage, because the standard regimen for adjuvant chemotherapy was not established for the enrolled patients at the time. However, the prognosis of patients with lymph node metastasis which was significantly correlated with THBS4-high might be improved.

To the best of our knowledge, the present study is the first to investigate the clinicopathological significance of THBS4 in the gastric cancer microenvironment. Macroscopic type 4 and the histologic diffuse type were significantly associated with a high expression of THBS4 in our patient series. Remarkably, THBS4 is expressed in most stroma of Borrmann's type 4 cancer. THBS4 is involved in tissue remodeling [6]. Macroscopic type 4, diffusely infiltrating carcinoma, and scirrhous gastric carcinoma (also known as linitis plastica-type carcinoma) are characterized by cancer cell infiltration and proliferation accompanied by extensive stromal fibrosis and abundant extracellular matrix [20]. This typical histological finding of macroscopic type 4 might be regulated in part by the tissue remodeling activity of THBS4.

FGFR2 is overexpressed on macroscopic type-4 gastric cancer [21]. The secretion of FGF7 (a ligand of fibroblast growth factor receptor 2 [FGFR]2) by gastric fibroblasts is likely to contribute in a paracrine manner to the remarkable cell proliferation seen in scirrhous gastric cancer with FGFR2 overexpression [22]. Whereas Huang et al. reported that the downregulation of THBS4 promotes the FGFR2 signal of gastric cancer progression via the PI3K-AKT-mTOR pathway using non-type-4 gastric cancer cell lines [23]. It is necessary in the future to examine the effects of THBS4 on the PI3K-AKT-mTOR pathway by using type-4 gastric cancer cell lines with FGFR2 overexpression.

In conclusion, THBS4 is expressed on CAFs in the microenvironment of gastric cancer, especially in macroscopic type-4 gastric cancer. THBS4 from CAFs might be associated with the invasion of cancer cells, and THBS4 is also a useful prognostic indicator for gastric cancer patients, especially for those with stage I or stage III cancer.

## Supporting information

**S1 Fig. The Kaplan-Meier survival curve for Borrmann's type 4 tumor and other type.** No significant difference in overall survival was shown between the THBS-high group and THBS-low group at Borrmann's type 4. Whereas the prognosis of the patients with high THBS4 expression were poorer than that of the patients with low THBS4 expression at other type. (TIFF)

**S1 File. Data of clinicopathologic factors in 584 patients.**
(XLSX)

**S1 Raw Images. Original images for blots and gels.**
(TIFF)

**S2 Raw Images. Original images for blots and gels.**
(TIFF)

**S1 Table. Correlation between the expression of THBS4 in tumor stromal cells and clinicopathologic features in Borrmann's type 4.**
(DOCX)

**S2 Table. Correlation between the expression of THBS4 in tumor stromal cells and clinicopathologic features in other macroscopic type.**
(DOCX)

**S3 Table. Recurrence pattern according to the stage.**
(DOCX)

## Author Contributions

**Conceptualization:** Kenji Kuroda, Masakazu Yashiro, Masaichi Ohira.

**Data curation:** Kenji Kuroda, Masakazu Yashiro.

**Formal analysis:** Kenji Kuroda, Masakazu Yashiro, Tomohiro Sera, Yurie Yamamoto, Yukako Kushitani, Atsushi Sugimoto, Syuhei Kushiyama, Sadaaki Nishimura, Shingo Togano, Tomohisa Okuno.

**Funding acquisition:** Masakazu Yashiro.

**Investigation:** Kenji Kuroda, Masakazu Yashiro, Tomohiro Sera, Yurie Yamamoto, Yukako Kushitani, Atsushi Sugimoto, Syuhei Kushiyama, Sadaaki Nishimura, Shingo Togano, Tomohisa Okuno, Masaichi Ohira.

**Methodology:** Kenji Kuroda, Masakazu Yashiro.

**Project administration:** Kenji Kuroda, Masakazu Yashiro, Masaichi Ohira.

**Resources:** Kenji Kuroda, Masakazu Yashiro, Tomohiro Sera, Yurie Yamamoto, Yukako Kushitani, Atsushi Sugimoto, Syuhei Kushiyama, Sadaaki Nishimura, Shingo Togano, Tomohisa Okuno, Tatsuro Tamura, Takahiro Toyokawa, Hiroaki Tanaka, Kazuya Muguruma, Masaichi Ohira.

**Software:** Kenji Kuroda, Masakazu Yashiro.

**Supervision:** Masakazu Yashiro, Masaichi Ohira.

**Validation:** Kenji Kuroda, Masakazu Yashiro.

**Visualization:** Kenji Kuroda, Masakazu Yashiro.

**Writing – original draft:** Kenji Kuroda, Masakazu Yashiro.

**Writing – review & editing:** Kenji Kuroda, Masakazu Yashiro, Masaichi Ohira.

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
