## [Decision Letter · Decision Letter 0]

6 Aug 2019

PONE-D-19-19270

The Clinicopathological Significance of Thrombospondin-4 Expression in the Tumor Microenvironment of Gastric Cancer

PLOS ONE

Dear Dr. Yashiro,

Thank you for submitting your manuscript to PLOS ONE. After careful consideration, we feel that it has merit but does not fully meet PLOS ONE’s publication criteria as it currently stands. Therefore, we invite you to submit a revised version of the manuscript that addresses the points raised during the review process.

(1) Immunostaining of Podoplanin (PDPN) as a widely accepted marker of cancer associated fibroblasts (CAFs)

(2) Detailed methodological description of CAFs and normal-associated fibroblasts (NAFs) and presentation of their images

(3) Explanation about THBS4 expression in stroll cells but not cancer cells in primary gastric cancer, as well as presentation of more clear images of THBS4 immunostaining, and results of THBS4 immunostaining

(4) Presentation and statistical analysis related to Western blot analysis

(5) Refinement of Kaplan-Meier analysis

(6) Additional experiments or discussion on the functions of thrombospondin-4 (THBS4) in CAFs

(7) Other issues pointed out by Reviewers

We would appreciate receiving your revised manuscript by Sep 20 2019 11:59PM. To enhance the reproducibility of your results, we recommend that if applicable you deposit your laboratory protocols in protocols.io, where a protocol can be assigned its own identifier (DOI) such that it can be cited independently in the future. For instructions see: http://journals.plos.org/plosone/s/submission-guidelines#loc-laboratory-protocols

We look forward to receiving your revised manuscript.

Kind regards,

Masaru Katoh, M.D., Ph.D.

Academic Editor

PLOS ONE

Journal Requirements:

This study is partially founded by KAKENHI Grant-in-Aid for Scientific Research, Nos. 18H02883 (M.Y.).

Additionally, because some of your funding information pertains to [commercial funding//patents], we ask you to provide an updated Competing Interests statement, declaring all sources of commercial funding.

In your Competing Interests statement, please confirm that your commercial funding does not alter your adherence to PLOS ONE Editorial policies and criteria by including the following statement: "This does not alter our adherence to PLOS ONE policies on sharing data and materials.” as detailed online in our guide for authors  http://journals.plos.org/plosone/s/competing-interests.  If this statement is not true and your adherence to PLOS policies on sharing data and materials is altered, please explain how.

Please include the updated Competing Interests Statement and Funding Statement in your cover letter. We will change the online submission form on your behalf.

Reviewers' comments:

Reviewer's Responses to Questions

**Comments to the Author**

1. Is the manuscript technically sound, and do the data support the conclusions?

Reviewer #1: Partly

Reviewer #2: Partly

Reviewer #3: Yes

Reviewer #4: Partly

2. Has the statistical analysis been performed appropriately and rigorously? 

Reviewer #1: Yes

Reviewer #2: Yes

Reviewer #3: Yes

Reviewer #4: Yes

3. Have the authors made all data underlying the findings in their manuscript fully available?

Reviewer #1: Yes

Reviewer #2: No

Reviewer #3: Yes

Reviewer #4: Yes

4. Is the manuscript presented in an intelligible fashion and written in standard English?

Reviewer #1: Yes

Reviewer #2: Yes

Reviewer #3: Yes

Reviewer #4: Yes

5. Review Comments to the Author

Reviewer #1: This manuscript is to evaluate that THBS4 expressed in CAFs is a prognostic marker to predict poor prognosis in gastric cancer. They proved that THBS4 was expressed in stromal cells expressing alpha-SMA in gastric cancer, and positive THBS4 expression is correlated with poor prognosis of gastric cancer patients. The results are clear, but for publication in Plos One, following points should be fixed.

1. Authors determined the expression of THBS4 in gastric cancer tissues according to the intensity and the percentage of stained cancer cells and stromal cells (Page 4, Line 71). However, they said that it was only expression in stromal cells. How did authors calculate the percentage of positive THBS cells? Was the percentage meaning the ratio of positive stromal cells in all cells including cancer cells and stromal cells?

2. In figure 1, the positive staining of THBS4 look like too diffuse in the entire tissues. Can authors show the clearer photo for THBS4 staining?

3. The gastric cancer generally showed histologic heterogeneity according to the classification. In figure 1, I recommend that authors show the representative photos for THBS4 expression in two different histological subtypes (diffuse and intestinal type).

4. In figure 1, the staining photo of cytokeratin, cancer cell specific maker, should be added to support that THBS4 was not stained in cancer cells.

5. In figure 3, authors just showed the expression of THBS4 in NAF and CAF. I recommend authors that the results should be added the expression of THBS of several gastric cancer cell lines.

6. Authors described in abstract and conclusion that “THBS4 from CAFs is associated with the metastasis of cancer cells” and “THBS4 from CAFs might be associated with the invasion of cancer cells”. However, they did not show the any data for proof of concept. Authors should add the experimental data to prove them, or the descriptions in abstract and conclusion should be toned down.

7. Authors emphasized that THBS4 is especially expression on CAFs of macroscopic type-4 gastric cancer. It would be better to add the results of subgroup analysis for type-4 or other type gastric cancer.

8. Authors insisted that non-significance of stage II would be explained as recurrent pattern and adjuvant chemotherapy. To supper this hypothesis, recurrent pattern or adjuvant regimen according to the stage should be added.

Reviewer #2: The authors investigated the clinicopathological significance of stromal thrombospondin-4 expression in the tumor microenvironment of gastric cancer, and found that THBS4 was expressed on CAFs. Stromal THBS4 was associated with the metastasis of cancer cells. The results are interesting and promising. However, this article suffers from several major flaws:

1. The authors used α-SMA to indicate the location of CAFs. However, α-SMA is a microfilament protein with contractile ability, mainly expressed at myoepithelial cells. Although α-SMA expression could also appear in the transformation of fibroblasts to myofibroblasts around cancer cells, the positive rate of α-SMA in CAFs is not high, and some of muscle tissues may be unexpectedly positive. I recommend the authors to add podoplanin (PDPN) in immunostaining assays, which is a widely accepted marker for CAFs with a relative high expression level.

2. Primary culture was performed to obtain CAFs and NAFs in this research. However, there was no demonstrations or photos to show the status of cells in order to verify the results. And the isolations and purifications of CAFs and NAFs were not mentioned either.

3. Inconsistent data were found between abstract and main text (e.g. positive rate of THBS4 expression).

4. For WB assay, the authors did not elucidate the sample size and necessary statistical analysis (t-test). A solitary result cannot clarify the real world.

5. According to the clinical data, THBS4 in stroma has obvious correlations with tumor invasion and lymph node metastasis. Since the primary CAFs have been cultured successfully, the authors should perform gene function experiments (e.g. migration & invasion) subsequently and explore the potential mechanisms to explain the phenomena they observed.

6. Are there any references to support the IHC evaluating method of THBS4 expression? If yes, please cite the articles, otherwise the authors should state the reason why score 3 was the watershed.

7. Terms were not used properly in this article. NAF was not common in the description of normal fibroblast. And definitions of positive and high were confused.

Reviewer #3: Dear editors,

In this study, the authors mainly investigated the clinicopathological value of THBS4

In gastric cancer based on the retrospective cohort from their own center. The research methods were statistical analyses of clinical data, including chi-square test, Cox proportional hazards regression models, as well as survival curves. The endpoint parameter was overall survival (OS). The article was well written, and focused on practical clinical aspects. While some problems still existed and should be improved.

1. The exact correlation of α-SMA or CAFs and THBS4 should be explained more apparently both in Introduction and Discussion.

2. The qualities of all K-M curves were poor. We believe this problem should be focused since key results of this study were reflected herein. Some normative forms could be referenced, for instance, ANNALS OF ONCOLOGY.

3. Information from Figure 3 was too simple to explain exact relation between the expression of THBS4 and α-SMA or CAFs.

4. References should be updated by using articles in recent 5 years.

Reviewer #4: The authors Kuroda et.al have identified THBS4 as a marker of CAFs in gastric cancer. IHC analysis on 584 GC patients specimen shows statistical significance when THBS4 expression is compared with the clinico-pathological characteristics of the patients. However, I have some comments with respect to the methodology of CAFs and NAFs isolation for western blot analysis. The authors have not elaborated on the specific markers that were used to identify these from the samples.

6. PLOS authors have the option to publish the peer review history of their article (what does this mean?). If published, this will include your full peer review and any attached files.

Reviewer #1: No

Reviewer #2: No

Reviewer #3: No

Reviewer #4: Yes: Kakoli Das

---

## [Author Response · Author response to Decision Letter 0]

26 Sep 2019

Reviewer #1

Thank you very much for the careful review of the Reviewer #1. We corrected several points according to the descriptions by the Reviewer #1, as described below. We indicated the changes point by point and highlighted them in the revised paper.

1. Authors determined the expression of THBS4 in gastric cancer tissues according to the intensity and the percentage of stained cancer cells and stromal cells (Page 4, Line 71). However, they said that it was only expression in stromal cells. How did authors calculate the percentage of positive THBS cells? Was the percentage meaning the ratio of positive stromal cells in all cells including cancer cells and stromal cells?

 We calculated the percentage of positive THBS cells by the ratio of stained stromal cells in all stromal cells. We added the comments in the materials and methods. (on page4 line 22-26)

2. In figure 1, the positive staining of THBS4 look like too diffuse in the entire tissues. Can authors show the clearer photo for THBS4 staining?

 We replaced the picture with the clearer photo for THBS4 staining in Figure 1. 

3. The gastric cancer generally showed histologic heterogeneity according to the classification. In figure 1, I recommend that authors show the representative photos for THBS4 expression in two different histological subtypes (diffuse and intestinal type).

 We added the picture of THBS4 expression in two different histological subtypes including diffuse type and intestinal type. We added some comments in the manuscript. (on page 7 line 12-13)

4. In figure 1, the staining photo of cytokeratin, cancer cell specific maker, should be added to support that THBS4 was not stained in cancer cells.

We evaluated the expression of cytokeratin as cancer cell specific maker in the gastric cancer tissues. Cancer cells were positive for cytokeratin staining, but negative for THBS4 staining. The pictures of cytokeratin were added in Figure 1. We added these comments in the manuscript. (on page 2 line 15) (on page 4 line 17) (on page 7 line 5-6) (on page 7 line 12) (on page 7 line 15)

5. In figure 3, authors just showed the expression of THBS4 in NAF and CAF. I recommend authors that the results should be added the expression of THBS4 of several gastric cancer cell lines.

In accordance with the Reviewer’s comment, we added western blot of THBS4 of two gastric cancer cell lines, MKN45 and NUGC4 in Figure 3B. We added some comments in the manuscript. (on page 2 line 13) (on page 2 line 17) (on page 5 line 9-10) (on page 11 line 12) (on page 11 line 18) (on page 12 line 3-7)

6. Authors described in abstract and conclusion that “THBS4 from CAFs is associated with the metastasis of cancer cells” and “THBS4 from CAFs might be associated with the invasion of cancer cells”. However, they did not show the any data for proof of concept. Authors should add the experimental data to prove them, or the descriptions in abstract and conclusion should be toned down.

We showed in the results that THBS4-high expression in stroma was significantly correlated with higher tumor invasion (p<0.001), lymph node metastasis (p<0.001), lymphatic invasion (p<0.001), peritoneal cytology (p<0.001), and peritoneal metastasis (p<0.001). Taken together, we found that the prognosis of patients with high THBS4 expression in tumor stroma was significantly poorer than that of patients with low THBS4 expression. These findings indicated that THBS4 from CAFs might be associated with the invasion and metastasis of cancer cells. We added these comments in the discussion. (on page 12 line 12-20)

7. Authors emphasized that THBS4 is especially expression on CAFs of macroscopic type-4 gastric cancer. It would be better to add the results of subgroup analysis for type-4 or other type gastric cancer.

We performed the subgroup analysis of type-4 or other types gastric cancer. No significant difference of clinic-pathological factors was found between THBS4-high group and THBS4-low group in Bormann’s type 4 cancer (S1 Fig and S1 Table). In contrast, the other types gastric cancer had significant differences between the two groups. (S1 Fig and S2 Table). THBS4 might be associated with Bormann’s type 4 cancer, but not associated with malignant progression of Bormann’s type 4 cancer. We added these comments in the manuscript. (on page 7 line 25-28) (on page 9 line 6-7)

8. Authors insisted that non-significance of stage II would be explained as recurrent pattern and adjuvant chemotherapy. To supper this hypothesis, recurrent pattern or adjuvant regimen according to the stage should be added.

 As reviewer pointed out, we added S3 Table which shows recurrence site according to the stage. Hematogenous recurrence was frequent in patients at stage II patients, in compared with those at stage III. In contrast, it was difficult to compare adjuvant regimen according to the stage, because the standard regimen for adjuvant chemotherapy was not established for the enrolled patients at the time. We added these comments in the text. 

(on page 9 line 10; on page 9 line 13)

Reviewer #2

Thank you very much for the careful review of the Reviewer #2. We correct several points according to the descriptions by the Reviewer #2, as follows.

1. The authors used α-SMA to indicate the location of CAFs. However, α-SMA is a microfilament protein with contractile ability, mainly expressed at myoepithelial cells. Although α-SMA expression could also appear in the transformation of fibroblasts to myofibroblasts around cancer cells, the positive rate of α-SMA in CAFs is not high, and some of muscle tissues may be unexpectedly positive. I recommend the authors to add podoplanin (PDPN) in immunostaining assays, which is a widely accepted marker for CAFs with a relative high expression level.

I agree with the Reviewer’s comment, we added the immunostaining experiment of podoplanin which evaluated the for CAF in tumor. The picture of podoplanin staining was added in Fig 1. And we commented the Podoplanin immunostaining in the text. (on page 2 line 15) (on page 4 line 11-20) (on page 7 line 8) (on page 7 line 15) (on page 12 line 5).

2. Primary culture was performed to obtain CAFs and NAFs in this research. However, there was no demonstrations or photos to show the status of cells in order to verify the results. And the isolations and purifications of CAFs and NAFs were not mentioned either.

We added representative pictures of CAFs and NFs by Podoplanin staining in Fig 3A. The method for the primary culture of CAFs and NFs was added in the manuscript. CAFs and NFs were defined by αSMA staining, as previously reported. (on page 5 line 4-9) (on page 5 line 12-16)

1. Fuyuhiro Y, Yashiro M, Noda S, Matsuoka J, Hasegawa T, Kato Y, et al. Cancer-associated orthotopic myofibroblasts stimulates the motility of gastric carcinoma cells. Cancer Sci. 2012;103(4):797-805. 

2. Fuyuhiro Y, Yashiro M, Noda S, Kashiwagi S, Matsuoka J, Doi Y, et al. Myofibroblasts are associated with the progression of scirrhous gastric carcinoma. Exp Ther Med. 2010;1(4):547-51.

3. Inconsistent data were found between abstract and main text (e.g. positive rate of THBS4 expression).

 We corrected them. (on page 2 line 11)

4. For WB assay, the authors did not elucidate the sample size and necessary statistical analysis (t-test). A solitary result cannot clarify the real world.

We added statistical analysis of western blot of THBS4 in CAFs and NFs in Fig 3B. We added the comments in the results. (on page 11 line 9-12) (on page 11 line 17-18). 

5. According to the clinical data, THBS4 in stroma has obvious correlations with tumor invasion and lymph node metastasis. Since the primary CAFs have been cultured successfully, the authors should perform gene function experiments (e.g. migration & invasion) subsequently and explore the potential mechanisms to explain the phenomena they observed.

We agree that additional experiments on function of THBS4 would be valuable, as reviewer commented. We plan to investigate THBS4 function in CAFs, and submit as a new paper of in vitro experiment in the near future. Then, we added these comments in the discussion, as follows. Muppala S et al. reported that THBS4 is upregulated in response to TGFβ and promotes the effect of TGFβ on angiogenesis, mediating cell proliferation and survival [3]. Forester et al. showed THBS4 transcription on CAFs was stimulated by incubation with conditioned medium derived from human diffuse gastric cancer lines [4]. It was speculated that TGF-β derived from cancer cells stimulated THBS4 expression on CAFs resulting in cancer proliferation and progression. We added these comments in the text. (on page 12 line 23-29)

3. Muppala S, Xiao R, Krukovets I, Verbovetsky D, Yendamuri R, Habib N, et al. Thrombospondin-4 mediates TGF-beta-induced angiogenesis. Oncogene. 2017;36(36):5189-98.

4. Forster S, Gretschel S, Jons T, Yashiro M, Kemmner W. THBS4, a novel stromal molecule of diffuse-type gastric adenocarcinomas, identified by transcriptome-wide expression profiling. Mod Pathol. 2011;24(10):1390-403.

6. Are there any references to support the IHC evaluating method of THBS4 expression? If yes, please cite the articles, otherwise the authors should state the reason why score 3 was the watershed.

We cited a reference which supports our IHC evaluating method of Thrombospondin4 (THBS4), as described below. 

5. Huang T, Liu D, Wang Y, Li P, Sun L, Xiong H, et al. FGFR2 Promotes Gastric Cancer Progression by Inhibiting the Expression of Thrombospondin4 via PI3K-Akt-Mtor Pathway. Cell Physiol Biochem. 2018;50(4):1332-45. 

7. Terms were not used properly in this article. NAF was not common in the description of normal fibroblast. And definitions of positive and high were confused.

We corrected NAF to NF. We also corrected “positive” to” high”. (on page 2 line 13) (on page 7 line 8-9) (on page 11 line 8)

Reviewer #3: Dear editors,

In this study, the authors mainly investigated the clinicopathological value of THBS4

In gastric cancer based on the retrospective cohort from their own center. The research methods were statistical analyses of clinical data, including chi-square test, Cox proportional hazards regression models, as well as survival curves. The endpoint parameter was overall survival (OS). The article was well written, and focused on practical clinical aspects. While some problems still existed and should be improved.

1. The exact correlation of α-SMA or CAFs and THBS4 should be explained more apparently both in Introduction and Discussion.

THBS4-high expression in stroma was significantly associated with higher αSMA expression. Forester et al. showed THBS4 is expressed on CAFs which are stimulated by diffuse gastric cancer cells. These findings suggest that CAFs with THBS4 expression might be activated by diffuse gastric cancer cells. Moreover, multivariate analysis for overall survival showed that not αSMA expression on CAFs but THBS4 expression on CAFs was an independent prognostic factor. CAFs with THBS4 expression might be associated with malignant potential of gastric cancer cells. We added these comments in the text. (Table 1) (Table 2) (on page 3 line 14-16) (on page 4 line 28-29) (on page 7 line 22) (on page 10 line 8) (on page 12 line 7-8) (on page 12 line 12-20)

 Forster S, Gretschel S, Jons T, Yashiro M, Kemmner W. THBS4, a novel stromal molecule of diffuse-type gastric adenocarcinomas, identified by transcriptome-wide expression profiling. Mod Pathol. 2011;24(10):1390-403.

2. The qualities of all K-M curves were poor. We believe this problem should be focused since key results of this study were reflected herein. Some normative forms could be referenced, for instance, ANNALS OF ONCOLOGY.

In accordance with the Reviewer’s comment, we renew the Kaplan-Meier curve referencing ANNALS OF ONCOLOGY. Graph size and line style were changed and censors were added.

3. Information from Figure 3 was too simple to explain exact relation between the expression of THBS4 and α-SMA or CAFs.

We added the comments of Fig 3 more in detail, as follows. (on page 11 line 14-18)

Figure legends

Fig 3. The picture of CAFs and NFs stained with Podoplanin and the western blot analysis of THBS4. (A), Representative picture of Podoplanin staining. Podoplanin was mainly stained at the cytoplasm fibroblasts. The expression level of Podoplanin was higher in CAFs, in compared with that in NFs. (B). THBS4 expression. CAFs expressed THBS4, but NFs and cancer cells did not.

4. References should be updated by using articles in recent 5 years.

 We updated references using recent articles.

Reviewer #4

Thank you very much for the careful review of the Reviewer #4. We correct several points according to the descriptions by the Reviewer #4, as follows.

I have some comments with respect to the methodology of CAFs and NAFs isolation for western blot analysis. The authors have not elaborated on the specific markers that were used to identify these from the samples.

 We used the specific markers, Podoplanin, to identify CAFs and NFs in Figure 3A. We also added the comments in the materials and methods and the results. (on page 5 line 12-16)

---

## [Decision Letter · Decision Letter 1]

22 Oct 2019

The Clinicopathological Significance of Thrombospondin-4 Expression in the Tumor Microenvironment of Gastric Cancer

PONE-D-19-19270R1

Dear Dr. Yashiro,

We are pleased to inform you that your manuscript has been judged scientifically suitable for publication and will be formally accepted for publication once it complies with all outstanding technical requirements.

With kind regards,

Masaru Katoh, M.D., Ph.D.

Academic Editor

PLOS ONE

Additional Editor Comments (optional):

Reviewers' comments:

Reviewer's Responses to Questions

**Comments to the Author**

1. If the authors have adequately addressed your comments raised in a previous round of review and you feel that this manuscript is now acceptable for publication, you may indicate that here to bypass the “Comments to the Author” section, enter your conflict of interest statement in the “Confidential to Editor” section, and submit your "Accept" recommendation.

Reviewer #1: All comments have been addressed

Reviewer #2: All comments have been addressed

Reviewer #3: All comments have been addressed

2. Is the manuscript technically sound, and do the data support the conclusions?

Reviewer #1: Yes

Reviewer #2: Yes

Reviewer #3: Yes

3. Has the statistical analysis been performed appropriately and rigorously? 

Reviewer #1: Yes

Reviewer #2: Yes

Reviewer #3: Yes

4. Have the authors made all data underlying the findings in their manuscript fully available?

Reviewer #1: Yes

Reviewer #2: Yes

Reviewer #3: Yes

5. Is the manuscript presented in an intelligible fashion and written in standard English?

Reviewer #1: Yes

Reviewer #2: Yes

Reviewer #3: Yes

6. Review Comments to the Author

Reviewer #1: Authors revised manuscript according to reviewers' comments.

It is acceptable to be published in PlosOne

Reviewer #2: The authors answered most of the questions I raised, and modified the mistakes. I am satisfied with the revised version.

Reviewer #3: (No Response)

7. PLOS authors have the option to publish the peer review history of their article (what does this mean?). If published, this will include your full peer review and any attached files.

Reviewer #1: No

Reviewer #2: No

Reviewer #3: No

---

## [Editor Report · Acceptance letter]

29 Oct 2019

PONE-D-19-19270R1 

The Clinicopathological Significance of Thrombospondin-4 Expression in the Tumor Microenvironment of Gastric Cancer 

Dear Dr. Yashiro:

I am pleased to inform you that your manuscript has been deemed suitable for publication in PLOS ONE. Congratulations! Your manuscript is now with our production department. 

With kind regards,

on behalf of

Dr. Masaru Katoh 

Academic Editor

PLOS ONE